# Contiguous Gene Syndromes and Hearing Loss: A Clinical Report of Xq21 Deletion and Comprehensive Literature Review

**DOI:** 10.3390/genes15060677

**Published:** 2024-05-23

**Authors:** Maria Teresa Bonati, Agnese Feresin, Paolo Prontera, Paola Michieletto, Valeria Gambacorta, Giampietro Ricci, Eva Orzan

**Affiliations:** 1Institute for Maternal and Child Health—Institute for Maternal and Child Health “Burlo Garofolo”, 34137 Trieste, Italy; paola.michieletto@burlo.trieste.it (P.M.); eva.orzan@burlo.trieste.it (E.O.); 2Independent Researcher, 33059 Fiumicello Villa Vicentina, Italy; 3Medical Genetics Unit, S. Maria della Misericordia Hospital, 06129 Perugia, Italy; paolo.prontera@ospedale.perugia.it; 4Department of Medicine and Surgery, Section of Otorhinolaryngology, University of Perugia, 06129 Perugia, Italy; gambacortavaleria@gmail.com (V.G.);

**Keywords:** hearing loss, congenital hearing loss, inner ear malformation, IP3, contiguous gene syndrome, microduplication, microdeletion, microarray, Xq21 deletion

## Abstract

Given the crucial role of the personalized management and treatment of hearing loss (HL), etiological investigations are performed early on, and genetic analysis significantly contributes to the determination of most syndromic and nonsyndromic HL cases. Knowing hundreds of syndromic associations with HL, little comprehensive data about HL in genomic disorders due to microdeletion or microduplications of contiguous genes is available. Together with the description of a new patient with a novel 3.7 Mb deletion of the Xq21 critical locus, we propose an unreported literature review about clinical findings in patients and their family members with Xq21 deletion syndrome. We finally propose a comprehensive review of HL in contiguous gene syndromes in order to confirm the role of cytogenomic microarray analysis to investigate the etiology of unexplained HL.

## 1. Introduction

Hearing loss (HL), the most prevalent sensorial defect at birth and during childhood, may be due to an alteration of the mechanic conduction of hearing waves through the external and middle ear, to a cochlear or neural defect in the signal transduction, or both, as it happens in conductive HL, in sensorineural HL (SHL) or in mixed HL, respectively [1,2]. In syndromic HL, the deficit may be associated with other medical findings, including malformations, anomalies, or pathologic alteration in any organ or system. Of note, approximately 20% of children presenting with HL as the only initial clinical feature will subsequently be diagnosed with syndromic HL [3,4]. The implementation of universal newborn hearing screening and audiologic surveillance make an early diagnosis of HL possible, optimizing patients’ management and allowing for prompt treatments [5]. Subsequent to the identification of HL, a detailed clinical and genetic work-up follows in order to reach the etiological diagnosis and to offer personalized care and follow-up to the patients and their families. These points are crucial especially in cases in which concurrent disability is present, as that may affect the outcomes.

Genetic causes are identified in almost 80% of prelingual HL cases [3], and more than 600 different syndromic conditions are known [6].

SHL is mostly the consequence of pathogenic variants (SNVs, Single Nucleotide Variants or indels) in single genes, while Copy Number Variations (CNVs) are an underestimated cause of non-syndromic HL. Indeed, pathogenic deletions are expected to occur sporadically across at least 100 HL-causing genes [7]. Current data show that sequence analyses of 100 HL genes together with the assessment of CNVs can identify a genetic cause in roughly one out of two patients with suspected hereditary HL, whereas ~20% of the patients with an etiopathogenetic diagnosis carry at least a causative CNV. However, only a few studies, mainly single-case or small cohort reports, have focused on contiguous gene syndromes exhibiting HL in association with congenital anomalies, neurodevelopmental disorders, and various syndromic conditions [8,9,10].

We report on a young boy with congenital, severe SHL associated with an inner ear malformation, who developed a neurodevelopmental disorder with autistic features, and choroideremia consistently with the detected novel hemizygous deletion of 3.7 Mb in Xq21.1q21.2. We therefore propose an extensive review of patients and families with Xq21 deletion syndrome and we suggest HL as a possible early-onset feature of contiguous gene syndromes. In addition, we propose to hold gross DNA rearrangements in consideration as a possible cause of HL.

## 2. Materials and Methods

### 2.1. Clinical Report

The clinical report summarizes information from a complete collection and revision of clinical and molecular documentation. Next Generation Sequencing (NGS) multi-gene panels were performed at R&I, and CGH arrays took place at the Genetic Laboratory of Perugia Hospital. Molecular analysis was performed in trio (proband and biological parents) from DNA extracted by peripheral blood. Multidisciplinary evaluations, including audiological, logopedic, neurological/NPI, cardiological assessments, inner ear imaging, abdominal US, brain MRI, and clinical follow-up, were performed at IRCCS Burlo Garofolo in Trieste.

### 2.2. Literature Review

(1)Xq21 deletion syndrome

A collection of reported cases was selected from OMIM [11] and Pubmed [12], and data were extrapolated and compared. A summary of the main findings is presented.

(2)Contiguous gene syndromes with HL

We researched and selected recent papers providing a comprehensive list of contiguous gene syndromes, microdeletion and microduplication syndrome in humans, and data merging. Searches of phenotype description of the selected syndromes took place on OMIM [11]; Orphanet [13]; Genereview [14]; and Pubmed [12] with a focus on the presence and characteristics of HL. A summary of the main findings is presented.

## 3. Clinical Report

The child was referred to our center at 3 years of age for a second opinion regarding a delay in obtaining positive results from speech therapy and psychomotricity.

### 3.1. Family History

The proband is an only child, born at term from a pregnancy obtained by heterologous intracytoplasmic sperm injection (ICSI), with oocytodonation, which had an unremarkable course. The father suffers from ischemic stroke consequences in essential thrombocythemia. There is no known family history of hearing deficits.

### 3.2. Clinical Presentation

The proband’s newborn hearing screening testing with otoacoustic emission failed bilaterally (“REFER”), and the second level audiologic evaluation (including tympanometry, diagnostic distortion product otoacoustic emissions, and click-evoked auditory brainstem response (ABR) testing) revealed a bilateral SHL, with an estimated hearing threshold of 60–70 dB HL in the right and 70–80 dB HL in the left ear. At 4 months of age, the child was fitted with two hearing aids. As part of the clinical investigations indicated for permanent congenital HL, a temporal bone CT revealed a bilateral inner ear malformation characterized by anomalous fusion of the internal auditory canal and the basal cochlear turn, a cochlea with an absent modiolus, the presence of the interscalar septa, and bulbous dilatation of the fundus of the internal auditory canal as shown in Figure 1a. Incomplete partition type III (IP3) was classified according to the Sennaroglu’s revised classification [15]. Absence of the modiolus of the cochlea and bulbous dilatation of the fundus of the internal auditory canal was confirmed by MRI. Brain MRI was normal.

The cardiological assessment, ECG, echocardiography, abdominal ultrasound, and ophthalmological examination performed at 2 years of age were all reported as unremarkable. The parents reported a motor delay with late achievement of milestones: he reached the sitting position without support at 11 months, crawled at 18 months, and achieved independent walking at 28 months, initially with a wide base and very unstable gait. Because of the impairment of global and gross motor skills and the presence of bilateral pes planum, he was treated with alternating orthotics and underwent physiotherapy and neuro-psychomotricity programs. Sphincter continence was also significantly delayed. Speech and language development was severely compromised with delays in comprehension, very limited verbal expression, and several phonemic errors, despite early speech therapy and neuropsychotherapy.

Overall, the audiological evaluation carried out at 3 years and 2 months of age confirmed a severe bilateral SHL associated with inner ear malformation. Although hearing abilities were not easily assessed due to the presence of attention difficulties, the hearing aid fitting provided a satisfactory amplified hearing threshold (30 dB HL). In contrast, motor, communication, and language delays were significantly worse than expected, considering the early amplification and rehabilitation. The following neurological consultation documented the presence of stereotypies (e.g., rocking of the head and flickering with the hands), poor gaze engagement, impulsive-oppositional behavior, and limited attention. The diagnosis was of Mixed Specific Developmental Disorder and attention deficit disorder. At the neurological follow-up, a neurodevelopmental disorder with global psychomotor delay and autism spectrum disorder (ASD) was diagnosed. At 3 years and 2 months of age, weight and height were approximately in the 30th percentile with OFC about 53.5 cm (unreliable measurement due to poor cooperation) (98th pc). He showed dolichocephaly, a prominent forehead, bilateral epicanthus, periorbital fullness, long eyelashes, and a broad nasal tip with anterverted nostrils. Mild dorsal hypertrichosis and keratosis pilaris on the arms and thighs were also evident, together with flat feet.

### 3.3. Genetic Analysis

Following clinical genetic evaluation, the NGS panel analysis of genes associated with congenital HL suggested a deletion in the *POU3F4* gene. Molecular karyotyping was then performed for confirmation: the CGH array revealed a hemizygous deletion of approximately 3.7 Mb in the chromosomal region Xq21.1q21.2, between the nucleotides 81464567 and 85234239 (release GRCh37), encompassing *POU3F4*, *CYLC1*, *RPS6KA6*, *HDX*, *APOOL*, *SATL1*, *ZNF711*, and *POF1B* and partially (exons 4–15) involving *CHM*. The deletion was inherited from the oocyte donor.

### 3.4. Ophthalmological Follow-Up

Following a reverse phenotyping approach, ophthalmological follow-up performed at 4 years of age showed a bilateral widespread pigment clumping in the middle retinal periphery. Snellen visual acuity was performed binocularly only, in accordance with age, and the kinetic binocular visual field resulted normal for age (visual acuity was 20/32) with a slightly hyperopic refraction, and with normal bilateral temporal extension (of 70 degrees). At the age of 5 years, visual acuity and the visual field were still normal. The electroretinogram (full-field ERG) showed a significant reduction in the scotopic component (30 micronvolts) and a modest reduction in the photopic one (60 micronvolts; a and b waves’ latencies were 21 ms and 43 ms, respectively). Finally, at the age of 6, areas of chorioretinal atrophy were found at the fundus, bilaterally in the middle retinal periphery, as well documented by retinography, shown in Figure 1b.

## 4. Xq21 Deletion Syndrome: Literature Review

At least 29 probands (28 males and 1 female) belonging to 14 unrelated families of different ancestries have been reported with syndromic association due to Xq21 deletion before our patient. Except for a de novo deletion and a patient whose mother was not genotyped, all the Xq21 deletions were maternally inherited.

We revised the molecular and clinical findings of all the affected patients (30) and their carrier mothers, when available. Carrier females (24) were evaluated following the molecular diagnosis in their son(s). A well-described cohort of heterozygous females has been also considered [16]. Clinical findings of the patients and the carrier females are summarized in Figure 2, which was drawn from the data reported in Appendix A. In the table, the breakpoints of the deletions are also shown.

All the reported probands had a major hearing issue: they all present bilateral sensorineural or mixed HL, of variable degree varying from moderate to profound, rarely asymmetric, and without documentation of progression according to the available follow-up. HL diagnosis was made during childhood and assumed to be “likely congenital” in some cases. Our proband is the only one in which HL is surely congenital, since the diagnosis was made after failing newborn hearing screening. All patients who underwent temporal bone imaging presented inner ear malformation, generally consistent with an IP3, as the here-reported patient. In addition, one patient underwent left-middle-ear exploration and stapedectomy, showing fixation of the stapes to the foot plate with a patent cochlear aqueduct and experienced marked perilymphatic gusher on stapedectomy, supporting DFNX2. Despite early hearing aid fitting and treatment, verbal comprehension and expression are significantly compromised, as we observed in our patient. Except for some patients, global developmental delay is a frequent concomitant feature with generally worse speech than motor development and a variable degree of intellectual disability, from moderate to profound. Hypotonia has also been reported in some patients (4). Behavior anomalies have been more rarely mentioned, with one patient presenting with psychotic behavioral and relational disorder and another showing severe social disability. In our patient, the comorbidity with attention deficit disorder and ASD probably contributed to poor verbal outcomes. Ophthalmologic signs and symptoms consistent with a diagnosis of choroideremia are present in a minority of children, and in most of the patients, it has been described as appearing after childhood. Facial phenotype, when reported (6), is not specific and clinically recognizable. Flat feet are present in our patient and in two other siblings (3) [17]. Among other clinical reported features, asymptomatic renal pelvic dilatation, a single microematuria episode, vescicoureteral reflux grade V, brain anomaly with mega cisterna magna, cerebral cysticercosis, and vestibular problems have been reported once, whereas obesity, short stature, and strabismus are rare findings.

In the cohort, a female proband exhibited bilateral HL, IP3, and developmental delay; no information about fundus oculi was provided [18,19]. Heterozygous females are generally asymptomatic, with normal hearing and vision. However mild ophthalmological signs including choroidal atrophy and retinal pigmentary stippling have been documented, as well as bilateral sensorineural mild HL [16,18] (Figure 2).

## 5. Contiguous Gene Syndromes with HL: Literature Review

The phenotypes of 192 recurrent and non-recurrent microdeletion and microduplication syndromes listed by Wetzel and Darbro [20], and 99 microdeletion and microduplication syndromes reported and described by Nevado et colleagues [9], have been revised with special focus on the presence, prevalence, and characteristics of HL. The clinical description of diseases mainly relies on information available in the free databases OMIM [11], Orphanet [13], and Genereview [14] and in papers in Pubmed [12], which could not be comprehensive of the totality of contiguous gene syndromes with HL. In particular, very rare and sporadic microdeletion or microduplications presenting with HL have not been included in our review.

Starting from over 200 microdeletion/microduplication syndromes [9,20], we extrapolated 56 imbalance-sensitive genomic loci associated with more than 60 syndromes, which may include HL as a clinical finding. Conductive, sensorineural, and mixed HL may occur commonly, occasionally, or rarely in the contiguous gene syndrome. In Table 1, Table 2 and Table 3, we focus on HL characteristics and HL candidate or responsible genes, whereas we provide only a summary of the main clinical features (column “Phenotype”) of the identified microdeletions/microduplication syndromes.

Few cases of genomic regions sensitive to microduplications have been associated with HL (4), as displayed in Table 2.

Finally, HL may present in both the reciprocal microdeletion/microduplication syndromes (11) for some recurrent genomic loci (7), displayed in Table 3.

## 6. Discussion

### 6.1. Xq21 Deletion Syndrome: Literature Review

We revised 22 clinical histories spanning over forty years and described a novel proband [17,18,51,52,53,54,55,56,57,58,59,60]. By reporting the phenotype exhibited by males in a three-generation family, i.e., choroideremia, obesity, and congenital deafness, Ayazi and colleagues suggested the existence of a new X-linked syndrome [55]. The pathomechanism underlying the condition present in this family and in a similar one, i.e., submicroscopic deletions in Xq21, was subsequently identified by Nussbaum RL [51]. Despite the limits of available technologies, it was possible to map the critical region for choroideremia and HL between DXYS1 and DXS72 markers [51,52,53]. The technological optimization for the deletion breakpoint determination in the first family was later published together with the family’s clinical follow-up and the description of other similar patients [52,56]. However, only with the advent of MLPA and microarray technologies (CGH or SNP) was it possible to define the breakpoints of the Xq21 deletions and reveal that its size varies from 5.2 [17] to 16 Mb [57]. Our patient is the first example of Xq21 deletion identified through a multi-gene NGS panel after neonatal diagnosis of an apparently isolated HL. Excluding the patient described by Song et al. [57] affected by HL only as carrier of a 1–1.5 Mb microdeletion at about 90 kb upstream of *POU3F4* (Appendix A), with a size of 3.7 Mb, it is the smallest Xq21 deletion and the only among those characterized deleting part of *CHM* (exons 4–15), whereas *POU3F4*, *CYLC1*, *RPS6KA6*, *HDX*, *APOOL*, *SATL1*, *ZNF711*, and *POF1B* were fully encompassed. Indeed, the breakpoints of only 3 out of the 12 different Xq21 deletions (Appendix A) were characterized through array technologies (CGH or SNP): genes involved in our patient were fully encompassed.

*POU3F4* and *ZNF711* haploinsufficiency is causative of the expression of audiological and neurodevelopmental phenotypes, respectively.

Indeed, *POU3F4* (OMIM #300039) is implicated in nonsyndromic X-linked deafness-2 (DFNX2, DFN3OMIM #304400), characterized by sensorineural or mixed HL in association with IP3, cochlear hypoplasia, and/or stapes fixation. In males, HL begins prelingually and progresses over time. Females with pathogenic variants in *POU3F4* tend to be less severely affected, with rare postlingual, mild HL and very rarely with inner ear anomalies [16]. DFNX2 is also caused by deletions and insertions located upstream of the gene, in a putative regulatory element region. *POU3F4* in DFNX2 may be partially or completely deleted [61,62,63].

*POU3F4* encodes a transcription factor that restricts the proliferation and lineage potential of neural stem cells [63]. It plays a role in neurogenesis of the inner ear, mediating the inner radial bundle formation [64]. In mice models for DFNX2, a dysfunction of spiral ligament fibrocytes in the lateral wall of the cochlea, leading to reduced endocochlear potential, underlies the sensorineural loss. Considering the pathogenic mechanism, some authors have recently proposed to develop a therapeutic approach in male Pou3f4 -/y mice based on gene transfer in cochlear spiral ligament fibrocytes mediated by an adeno-associated viral (AAV) vector with a strong tropism for the spiral ligament (AAV7). Complementary gene replacement before HL progression to profound deafness could represent an attractive strategy to prevent fibrocyte degeneration and to restore normal cochlear functions and properties, including a positive endocochlear potential [65].

*ZNF711* (OMIM #314990) codes for the zinc-finger protein 711, ZNF711, the specific function of which is unknown. Some evidence suggests that it is crucial for brain development, probably acting as a transcription factor for genes required for neuronal development [66]. Nonsense, frameshift, and missense variants have been reported in *ZNF711*-related nonsyndromic intellectual developmental disorder, X-linked 97 (OMIM #300803), characterized by generally mild motor and severe speech developmental delay, mild-to-moderate intellectual disability in males, and autistic features in some of them. Despite some mild phenotypic features having been described, there are no distinctive facial dysmorphologies that would allow clinical recognition [67]. Female carriers do not show manifestations.

Most of the pathogenetic variants in *CHM* (OMIM #303100), associated with choroideremia, are nonsense, splicing, frameshift, and small intragenic deletion/duplications, resulting in the truncation or complete absence of Rab escort protein 1 (REP1) [68,69,70,71].

*POF1B* (OMIM #300603) has been proposed as a candidate gene for Premature ovarian failure 2B (300604) in female carriers, but controversial findings have been reported [72,73,74].

Although *HDX* and *RPS6KA6* are not OMIM disease genes, they are candidates for intellectual disability. For instance, two alterations (one duplication and one translocation) involving the gene and associated with X-linked intellectual disability and premature ovarian failure, respectively, have been reported in HGMD^®^ [75]. *RPS6KA6* encodes for a constitutively active kinase that belongs to the same family of a protein whose loss of function causes Coffin–Lowry syndrome, a syndromic intellectual disability with hypotonia. Moreover, at least two missense variants in *RPS6KA6* have been reported in association with psychomotor retardation, ASD and HL [75].

Although some authors suggested an age-dependent penetrance for choroideremia [17], in our patient, an early onset of the phenotype has been documented. As reported for females carriers of *POU3F4* pathogenic variants [16], Xq21 deletion carrier females can also exhibit mild ophthalmological signs including choroidal atrophy and retinal pigmentary stippling, as well as bilateral mild SHL and rarely inner ear anomalies. Moreover, even a female may exhibit the full spectrum of the Xq21 deletion syndrome, that includes neurodevelopmental delay (Figure 2) [16,18]. Furthermore, we underline that the identification of heterozygous females is crucial for reproductive counseling as well as for clinical evaluation and follow-up.

The phenotype of patients with contiguous gene syndrome may be more complex than the sum of specific signs and symptoms related to single genes. In particular, the co-occurrence of double sensorial deficit and neurodevelopmental disorder significantly impacts on prognosis, outcomes, and management choices in children with Xq21 deletion. Interestingly, neurodevelopmental issues have been frequently observed in children with *POU3F4*-related IP3, even in patients who are carriers of pathogenic variants [18].

### 6.2. Considerations on Etiologic Diagnosis of HL

Early identification of HL and consequent etiology determination impact the personalization of the patient’s follow-up, prognosis and outcome prediction, and in certain cases also the choice of treatments and therapeutic options. In syndromic HL, a multidisciplinary follow-up is required, according to the phenotype.

Following the diagnosis of *POU3F4*-related HL, imaging is mandatory in search of inner ear anomalies [61]. Cochlear implant surgery in IP3-III may be difficult due to the leakage of cerebrospinal fluid into the middle ear cavity when opening the round window, or performing the cochleostomy, with a prolonged “gusher” [76,77]. This increases the risk of further post-surgical cerebrospinal fluid leaks which can also lead to rhinorrhea, meningitis, and intracranial infections. A second surgical risk concerns the possibility of incorrect positioning of the electrodes in the internal acoustic canal due to the absence of the modiolus and the widening of the bottom of the internal acoustic canal [77]. Therefore, the choice of cochlear implant surgery requires careful counseling with the family and the sharing of therapeutic rehabilitation choices [78].

The identification of the causative genomic Xq21 deletion allowed for the indication of our patient specific follow-up and the early diagnosis of choroideremia. Choroideremia results from the progressive, centripetal loss of photoreceptors and choriocapillaris, secondary to the degeneration of the retinal pigment epithelium (RPE) [79]. Affected individuals present in late childhood or early teenage years with nyctalopia and progressive peripheral visual loss. Typically, by the fourth decade, the macula and fovea also degenerate, resulting in advanced sight loss [80,81]. In the first phases of this dystrophy, peripheral pigmentary changes may characterize the retina of affected patients [82]. At a later time, distinct regions of chorioretinal atrophy are usually visible. Of note, these degenerative alterations usually start at the equator and progressively, in a centripetal direction, involve the posterior pole and the peripapillary region [83]. These retinal changes can lead to the appearance of multiple scotomas in the peripheral visual field [84]. The full-field ERG may have reduced amplitude, initially in the scotopic component only, or be extinct [83,85]. Children under 7 years have rarely been diagnosed [82,84,85]. In our case, considering the initial alterations of the fundus as early as 4 years, the presence of areas of chorioretinal atrophy at 6 years, and an altered ERG at 5 years, which presented an initial involvement of the retinal cones, the ophthalmologic involvement is to be considered very serious with probable poor outcome and early visual loss. Close audiologic evaluations are obviously scheduled for the patient, also considering the possibility of hearing impairment progression. Therapeutic options have to consider the increased risk of complete loss of hearing in the case of middle ear surgery due to the identification of a specific inner ear anomaly.

In addition, the etiological diagnosis allows for the definition of the recurrent risk and the discussion of reproductive options in family counseling. In each pregnancy, the apparently healthy mother of a male proband has a 50% chance of transmitting the CNV loss; consequently, males who inherit the deletion will be affected, whereas females may be healthy or symptomatic. Furthermore, carrier mothers should undergo hearing and vision evaluations and follow-up. In the here-reported patient, challenges in genetic counseling emerged in both the donor and recipients of the oocyte, facing new ethic, legislative, and clinical needs.

### 6.3. HL in Contiguous Gene Syndromes

Unraveling the etiology of syndromic HL means we must consider the involvement of a single gene with a pleiotropic effect, or of two genes in double diagnoses as well as contiguous gene syndromes. Contiguous gene syndromes include genomic disorders with reciprocally deleted or duplicated chromosomal regions and microdeletion/microduplication syndromes. In these last ones, CNVs may randomly involve any genomic region, causing variable clinical presentations [86,87]. Of note, few contiguous gene syndromes may exhibit isolated HL, such as in female carriers of Xq21 deletion, in the frequent autosomal recessive 15q15.3 deletion syndrome in females, or in the very rare 9q21.11 duplication syndrome. We then focused on the pathomechanism or susceptibility factors for HL. Anatomical features, such as craniofacial abnormalities and ear malformations, immune deficit, and susceptibility to frequent ear infections, may contribute to conductive or mixed HL (Table 1, Table 2 and Table 3). Inner ear malformation has been reported not only in Xq21 microdeletion syndrome, in which HL and IP3 are common and specific findings are related to *POU3F4* haploinsufficiency, but also in 22q11.2 microdeletion syndrome, also known as DiGeorge syndrome, characterized by incomplete penetrance and an extremely variable phenotype (Table 3).

As shown in Table 1, Table 2 and Table 3, in nearly half of the contiguous gene syndromes, there are HL candidate genes. Some of them (13) are known to be causative of monogenic hereditary HL according to the deafness database [38]. We can therefore hypothesize that they may be involved in causing HL in the contiguous gene syndromes, where they are deleted or duplicated. For the remaining half of contiguous gene syndromes, the etiological mechanism of HL is unknown. We can only infer that dosage-sensitive genes mapped into the rearrangements might be candidates for HL. *COCH* is associated both with autosomal dominant nonsyndromic HL with variable penetrance of vestibular malfunction (DFNA9), Meniere’s disease, and possibly glaucoma and with autosomal recessive nonsyndromic HL. *SERPINB6*, *MYO15A*, and *OTOA* have been associated with autosomal recessive non-syndromic HL; thus, a possible second-hit event could be hypothesized in patients with the relative deletion syndromes exhibiting HL. *USH1C* may be associated both with autosomal recessive nonsyndromic HL and with Usher syndrome. Bilallelic deletions in 15q15.3 involving *STRC*/*CATSPER* are responsible for autosomal recessive nonsyndromic HL in females and autosomal recessive HL with asthenoteratozoospermia in males. HL may also be due to the presence of a pathogenic SNV in *STRC* in trans with the deletion. Finally, similarly to the disease mechanism related to *POU3F4*, *COL4A5* causes Alport syndrome both in case of a pathogenetic SNV or complete gene deletion.

### 6.4. The Role of Chromosomal Microarray in HL

Recently, the European Network for Genetic Hearing Impairment published recommendations for the evaluation of prelingual HL, underling the indication of chromosome assessment by a CGH array, investigating different genes located contiguously on a chromosome segment, in case of a polymalformative syndrome, or a set of clinical signs associated with deafness that does not evoke a known diagnosis [88]. With mostly genetic HL explained by monogenic conditions, isolated chromosomal microarray testing in an individual with apparent nonsyndromic HL has a low diagnostic yield, as stated in the dedicated GeneReviews page on the genetics of HL [89]. Congenital HL may actually be isolated in several cases, due to environmental causes such as CMV infection or due to a genetic cause, including the most frequently mutated *GJB2* gene or large numbers of other genes [3]. Since there are hundreds of syndromes presenting with congenital HL and characterized by additional signs and symptoms manifesting after birth, during childhood or later, caution should be posed in the definition of “nonsyndromic HL” at birth, in early infancy, and during childhood, especially before a definite etiologic diagnosis. With this in mind, the choice of comprehensive genetic analysis is nowadays preferred to HL multi-gene panels. In the starting era of neonatal genomic screening, exome sequencing has been proposed as an efficient first-tier analysis to screen for monogenic causes of congenital HL [90]. The promising benefits of early etiologic diagnosis by NGS strategies are arising, together with challenges in the interpretation of molecular data, related to the possibility of a partial phenotype at birth, variant of uncertain significance, and limited genotype–phenotype correlations. Whenever comprehensive NGS tests turn out to be nonconclusive, genomic disease mechanisms should be considered.

In contiguous gene syndromes, HL may be caused by the presence of dosage-sensitive gene(s) mapped in the CNV or gene disruption at the CNV breakpoint. Although chromosomal microarray testing in an individual with HL has a low diagnostic yield, we identified about 50 loci associated with almost 60 contiguous gene syndromes that could exhibit HL. As discussed in the previous paragraph, in some of them, we already know or can hypothesize the gene(s) involved in or that can cause HL.

Congenital HL identified by newborn hearing screening may be the first clinical feature in those microdeletion or microduplication syndromes exhibiting minor congenital anomalies, usually undetectable by prenatal imaging, such as inner ear malformation, neurodevelopmental disorder, and other sensorial defects. Indeed, only in a minority of the reviewed microdeletion or microduplication syndromes, such as Wolf–Hirschhorn syndrome (4p16.3 DS, Table 1), Cri-du Chat syndrome (5p15 DS, Table 1), and microdeletion or microduplications associated with split hand/foot malformation, do patients exhibit at birth a recognizable pattern of signs/multiple congenital anomalies. This is the subgroup that could be diagnosed in a prenatal setting [91].

According to the existing recommendations, the evidence, and this review, we point out that we should consider contiguous gene syndromes and perform CMA or locus-specific MLPA analysis in patients exhibiting HL whenever a deletion is suspected at NGS or only a monoallelic variant has been identified in patients in which a recessive syndrome is supposed, in evocative syndromic associations, and in undiagnosed patients with likely syndromic HL. Finally, we believe that this review may help geneticists, audiologists, and clinicians to counsel about HL in contiguous gene syndromes, especially in newborns.

## Figures and Tables

**Figure 1 genes-15-00677-f001:**
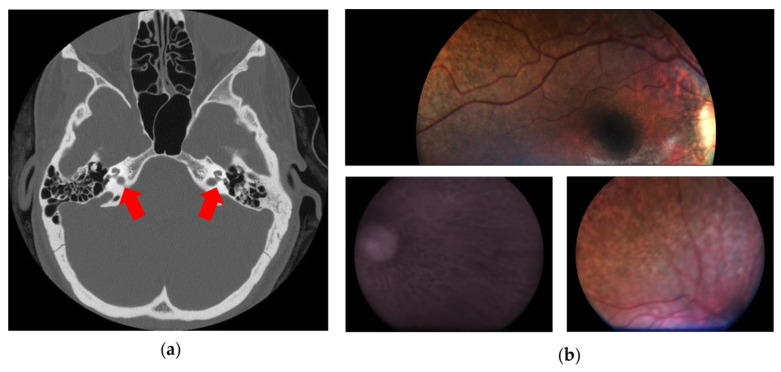
This figure shows the main clinical features of the patient. (**a**) Computed tomography of temporal bones (axial plane) showing bulbous dilatation at the distal ends of internal auditory canals; the interscalar septa of the cochlea are present, but the modiolus is absent (red arrows). (**b**) Fundus oculi showing choroidal degeneration and diffuse retinal pigmented epithelium dystrophy.

**Figure 2 genes-15-00677-f002:**
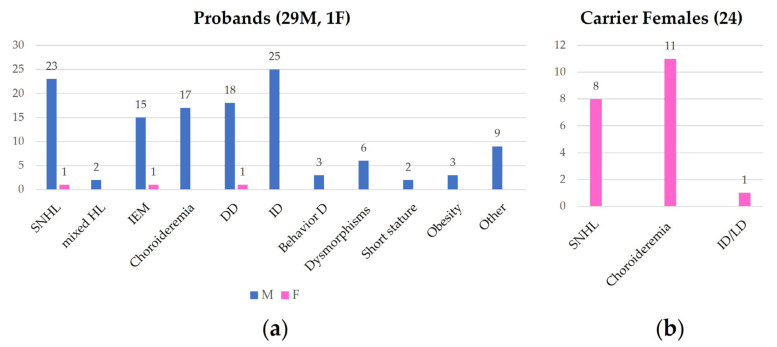
This figure shows the frequence of clinical findings in affected probands (**a**) and in carrier female family members (**b**). SNHL: sensorineural hearing loss; HL: hearing loss; IEM: inner ear malformation; DD: developmental delay; ID: intellectual disability; Behavior D: behavior disorder; LD: learning disability.

**Table 1 genes-15-00677-t001:** Microdeletion syndromes with hearing loss (HL).

Deletion Syndrome (DS)	Deletion Size, Genomic Coordinates (GRCh38)	HL Classification	HL Frequency	Phenotype	Gene/s Associated with HL °	Ref *
**Chr 1p36 DS, Distal (OMIM #607872)**	2.2 to 10.6 Mb terminal and interstitial dels throughout the 30 Mb of 1p36	SNHL	**35–82%**	Recognizable craniofacial features, DD/ID, hypotonia, epilepsy, cardiomyopathy/CHD, brain abnormalities, eyes/vision problem, short stature	*RERE*	
**Chr 1p35 DS** **(OMIM #617930)**	0.3 to 5.6 Mb	Monolateral, bilateral, mixed, SN	occ	IUGR, short stature, DD, LD, hypermetropia, recognizable facial dysmorphisms (prominent forehead, long, myopathic facies, fine eyebrows, small mouth, and micrognathia)	*HDAC1* hyp	[21]
**Chr 2p16.1-p15 DS (OMIM #612513)**	3.9 Mb	HL, SNHL	occ	DD, ID, behavioral disorders, recognizable dysmorphic features (microcephaly, bitemporal narrowing, smooth and long philtrum, hypertelorism, downslanting palpebral fissures, broad nasal root, thin upper lip, high palate), brain abnormalities (pachygyria, hypoplastic corpus callosum), asymptomatic persistence of HbF (if *BCL11A* deleted)	*USP34*, *XPO1*	[22]
**2p14-p15**	2.23–2.84 Mb, a SRO of 10 genes	HL, SNHL	occ	ID, speech disorder, mild dysmorphic features, relative microcephaly	*SPRED2*	
**Chr 2p12-p11.2 DS (OMIM #613564)**	Various dels (7.45–11.4 Mb)	SNHL	occ	DD with neurologic involvement, skeletal and genitourinary anomalies, dysmorphic features	unkn	
**Chr 2q37 DS (OMIM #600430) Albright Hereditary Osteodystrophy-Like S,** Brachydactyly-Mental Retardation S, BDMR; (overlap with Smith Magenis S)	MCR 200 kb (*HDAC4*) for BDMR	SNHL	occ	Characteristic facial dysmorphisms, DD/ID, hypotonia in infancy, abnormal behavior with ASD, brachydactyly type E, short stature, obesity	unkn	
**Chr 3pter-p25 DS** **(OMIM #613792) 3p- syndrome**	6–12 Mb	HL	occ	ID, motor DD, unusual facial features (microcephaly, micrognathia, ptosis, long philtrum, low and deformed ears, polydactyly deformity), hypotonia, CHD, renal and gastrointestinal malformations, autism, congenital hypothyroidism, epilepsy, tumors	unkn	
**Chr 3p14p12 DS** (overlap with CHARGE S)	interstitial chr 3p deletion encompassing the 3p14 band (*FOXP1*, *MITF*, *ATXN7*, *PRICKLE2*, *ROBO1*, and *ROBO2*)	HL, SNHL	occ	Global DD, neurological problems, failure to thrive, facial dysmorphisms, short stature, CHD, urogenital abnormalities, joint contractures were present in the older patients	*MITF* °, *FOXP1*	[23]
**Chr 3p13 DS *FOXP1*-related IDS**	Dels encompassing *FOXP1*	HL	occ	Moderate-to-severe DD, growth delay, HL, distinctive facial features (prominent forehead, mid facial hypoplasia)	*FOXP1*	[24]
**Chr 4p16.3 DS** **(OMIM #194190)** **Wolf-Hirschhorn S**	Dels: small del (<3.5 Mb); large del (5–18 Mb); very large del (>22–25 Mb); complex chr rearrangements MCR located at 4p16.3 (200–750 kb including *WHSC1*, *LETM1*, *NSD2*, *CPLX1*, *PIGG*)	Moderate-to-severe HL, mostly conductive, often secondary to chronic otitis media; SNHL, with and without conductive HL	**40%**	Pre- and postnatal growth retardation, hypotonia, ID, epilepsy, craniofacial dysmorphisms, congenital fusion anomalies	unkn	
**Chr 5p DS** **(OMIM #123450)** **Cri-du-chat S, Cat cry S**	560 kb in 5p15.2 to 40 Mb in 5p/entire 5p arm	Bilateral SNHL, auditory neuropathy (AN)	**com**	Distinctive facial features (microcephaly, round face, hypertelorism, micrognathia, epicanthal folds, low-set ears), hypotonia, severe DD, ID, characteristic high-pitched, monochromatic, “cat-like” cry	β-catenin	[25]
**Chr 5q14.3-q15 DS/5q14.3 DS Distal (OMIM #612881)**	Dels encompass *MEF2C*	HL	-	Severe DD, epilepsy or febrile seizures, muscular hypotonia, variable brain and minor anomalies. 5q14.3-q21 del associated with iris coloboma, HL, dental anomaly, moderate ID, ADHD	unkn	
**Chr 5q35 DS (OMIM #117550)** **Sotos S**	Dels encompass *NSD1*	Conductive HL associated with otitis media with effusion, cholesteatoma; high-frequency SNHL	15%	Distinctive facial appearance (broad and prominent forehead with a dolichocephalic head shape, sparse frontotemporal hair, downslanting palpebral fissures, malar flushing, long and narrow face, long chin), LD (early DD, mild-to-severe ID); overgrowth (height and/or head circumference ≥ 2 SD above the mean). Behavioral findings (ASD), advanced bone age, cardiac anomalies, cranial MRI/CT abnormalities, joint hyperlaxity, pes planus, renal anomalies, scoliosis, and seizures. Maternal preeclampsia, neonatal complications	unkn	
**Chr 6p25** **DS (OMIM #612582)** (overlap with Axenfeld-Rieger S type 3 and Branchiooculofacial S (*TFAP2A* in 6p24.3, centromeric to the del interval)	SRO of 1.3 Mb	SNHL	**com**	Hydrocephalus; white matter abnormalities; ocular, craniofacial, skeletal, cardiac, and renal malformations; bicuspid aortic valve; short stature; dysmorphic face	*SERPINB6* °	[26]
**Chr 6q24-q25** **DS (OMIM #612863)**	SRO of 3.52 Mb	Bilateral moderate-to-severe SNHL	**com**	IUGR, growth delay, ID, cardiac anomalies, facial dysmorphisms	*NOX3* hyp	
**Chr 7p22.1 DS (OMIM #243310)** **Baraitser-Winter S**	Dels including *ACTB*	HL, sensorineural. Mild conductive HL	occ	Typical craniofacial features and ID. Many affected individuals have frontal pachygyria, wasting of the shoulder girdle muscles, iris or retinal coloboma, and/or SRHL. Seizures, CHD, renal malformations, and GI dysfunction are also common	unkn	
**Chr 8q13.3 DS** **Branchio-oto-renal S**	Del encompassing *EYA1*	HL	**com**	Branchial defects, preauricular pits, renal anomalies	*EYA1* ° hyp	[27]
**Chr 8q21.11 DS (OMIM #614230)**	539.77 kb (range 0.66–13.55 Mb)	SNHL	occ	ID, hypotonia, decreased balance, unusual behavior, round face with full cheeks, a high forehead, ptosis, cornea opacities, underdeveloped alae, short philtrum, cupid’s bow of the upper lip, down-turned corners of the mouth, micrognathia, low-set and prominent ears, mild finger and toe anomalies (camptodactyly, syndactyly, and broadening of the first rays)	unkn	
**Chr 9q34.3 DS/9q- S/9q Subtelomeric DS (OMIM #610253) Kleefstra S 1; KLEFS1**	9q subtelomeric dels involving *EHMT1*	HL sensorineural and/or conductive, starting at a young age	**com**	ID, autistic-like features, childhood hypotonia, distinctive facial features, CHD, renal/urologic defects, genital defects in males, severe respiratory infections. The majority of individuals present moderate-to-severe spectrum of ID although a few individuals have mild delay and low-normal IQ. Severe expressive speech delay with little speech development, epilepsy/febrile seizures, psychiatric disorders, and extreme apathy or catatonic-like features after puberty	*EHMT1* hyp	
**Chr 10p14-p13 DS (OMIM #601362) DiGeorge syndrome/velocardiofacial syndrome complex-2 (DGS/VCFS 2)**	Dels at 10p14-p13/monosomy 10p	HL, sensorineural and/or conductive	**com**	CHD, immune deficiency, hypoparathyroidism, cleft palate, DD, microcephaly, cryptorchidism. Hemizygosity of the proximal region (DGCR2) can cause cardiac defect and T cell deficiency. Hemizygosity of the distal region (HDR1) can cause hypoparathyroidism, SNHL and renal dysplasia/insufficiency or a subset of this triad		
**Chr 10q26 DS (OMIM #609625)**	at least 600 kb in 10q26.2	SNHL, mild to moderate	**com**	Characteristic facial appearance, variable cognitive impairment, and neurobehavioral manifestations	unkn	
**Chr 11p11.2 DS (OMIM #601224)** **Potocki-Shaffer S**	del of 2.1 Mb in 11p12 p11.2	SNHL, mild to moderate	**com**	Craniofacial abnormalities, DD, ID, multiple exostoses, biparietal foramina	unkn	
**Chr 11p14-p15 DS**	122 kb involving *ABCC8* and *USH1C* *biallelic CGS*	Profound SNHL	**com**	Hyperinsulinism, enteropathy, renal tubular dysfunction	*USH1C* °	[28,29]
**Chr 11q13 DS (OMIM #166750) Otodental S or oculo-oto-dental S**	del of 43–490 kb	Severe SNHL (high frequency), progressive, onset from early childhood to middle age	**com**	Dental abnormalities and high-frequency SNHL, ocular coloboma in some cases. Severe deafness, microtia, and small teeth, without eye abnormalities, in association with a 2.75 Mb deletion in 11q13.2-q13.3	*FGF3* hyp	
**Chr 11q23-qter DS/Chr 11q DS/Partial 11q monosomy S (OMIM #147791) Jacobsen S**	from ~7 to 20 Mb, with the proximal bkp within or telomeric to 11q23.3 and the deletion extending usually to the telomere	HL, SNHL, mild hearing impairment	unkn	Pre- and postnatal physical growth retardation, DD, characteristic facial dysmorphisms (skull deformities, hypertelorism, ptosis, coloboma, downslanting palpebral fissures, epicanthal folds, broad nasal bridge, short nose, v-shaped mouth, small ears, low-set posteriorly rotated ears). Abnormal platelet function, thrombocytopenia or pancytopenia are usually present at birth. Patients commonly have malformations of the heart, kidney, GI tract, genitalia, central nervous system, and skeleton. Ocular, hearing, immunological, and hormonal problems may be also present	*ETS1*, *FLI1* hyp	[30]
**Chr 13q14 DS (OMIM #613884)**	at least 16 Mb encompassing 39 genes	HL	rare	Retinoblastoma, variable degrees of mental impairment, characteristic facial features, including high forehead, prominent philtrum, and anteverted earlobes	unkn	
** 13q21.2-q31.1 **	Del of 25.1 Mb, with bkps at D13S1289 and D13S886. The Del is the result of the unbalanced segregation (der(13)) of the insertional translocation 46,XY,ins(3;13)(p21.1;q22.3q32.1)	Deafness	occ	Duodenal stenosis, developmental and growth delay, vertebral anomalies, facial dysmorphisms	*KLHL1, CTD12/PFET1* hyp	[31]
**Chr 13q22 DS**	Del of the distal long arm of Chr 13	Profound SNHL	occ	Waardenburg-Shah syndrome: hypertelorism, flat nasal bridge, bright blue irises with elliptical pupils, Hirschsprung disease, anteriorly displaced anus, DD	*EDNRB* ° (AD Ws)	[32]
**Chr 14q11-q22 DS (OMIM #613457) Zahir-Friedman S**	Del from 3.0 to 40 Mb, without recurrent bkps	Mild hearing impairment	occ	When the del includes *FOXG1*, *NKX2-1*, and *PAX9*: severe ID, CNS malformations (corpus callosum agenesis). More distal deletions, involving *NKX2-1* and *PAX9*, appear to be associated with a milder phenotype	*COCH* °	
**Chr 14q22-q23 DS (OMIM #609640)** **Frias S**	4.06 Mb	HL, unilateral, mild	occ	Mild exophthalmia, palpebral ptosis, hypertelorism, short square hands with minimal proximal syndactyly between the second and third fingers, small broad great toes, short stature. Some patients may exhibit bilateral pedunculated postminimi	unkn	
**Chr 15q15.3 DS (OMIM #611102) Deafness-Infertility S**	*biallelic CGS*	HL, SNHL (bilateral, prelingual)	**very com**	Early-onset deafness in both males and females and exclusive male infertility (asthenoteratozoospermia)	biallelic *STRC* ° pathogenic variants/one *STRC* pathogenic variant and one contiguous gene deletion involving *STRC*	
**Chr 15q26-qter DS (OMIM #612626) Drayer S**	5.8 Mb encompassing *IGF1R*	HL, SNHL, bilateral	occ	Prenatal and postnatal growth retardation, microcephaly, DD, CHD	unkn	
**Chr 17p13.1 DS (OMIM# 247200) Miller-Dieker lissencephaly S**	180 kb (range 287 kb to 4.4 Mb) encompassing 18 genes, due to *PAFAH1B1* or *LIS1* haploinsufficiency	Conductive HL secondary to chronic otitis media	occ	CNS anomalies (subcortical band heterotopia, agyria/pachygyria or type I lissencephaly, ventriculomegaly, corpus callosum dysgenesis/agenesis), microcephaly, seizures, facial dysmorphisms (prominent forehead and occiput, bitemporal narrowing, furrowed brow, small nose, anteverted nostrils, low-set ears, prominent lip and micrognathia), hypoplastic male external genitalia, IUGR, ID and extracranial anomalies of cardiac defects, omphalocele and genitourinary abnormalities	unkn	
**Chr 17q21.31 DS (OMIM #610443) Koolen De Vries S**	600–800 kb del encompassing *CRHR1*, *MAPT*, *STH*, *IMP5*, *KANSL1*	HL, most commonly conductive, although SNHL has been reported	occ	DD/ID (in the mild-to-moderate range), neonatal/childhood hypotonia, speech and language delay (100%), dysmorphisms, behavioral features (friendly, amiable, cooperative). Other findings include epilepsy (~33%), CHD (25–50%), renal and urologic anomalies (25–50%), cryptorchidism	*KANSL1* hyp	
**Chr 17q22-q23.2 DS**	The deletions encompass *NOG*, *TBX2*, *TBX4*. The same *locus* is involved in the reciprocal dup, where HL is not reported	SNHL, bilateral	occ	Microcephaly, prenatal onset growth restriction, heart defects, tracheoesophageal fistula, esophageal atresia, skeletal anomalies, moderate-to-severe global DD	unkn	[9]
**Chr 18q DS/18q- S (OMIM #601808)**		HL, sensorineural, conductive. Congenital atresia or stenosis of the external ear canals associated with dels between D18S812 and D18S1141 (18q22.3-q23)	**com**	Short stature, ID, hypotonia; malformations of the hands and feet, abnormalities of the skull and craniofacial region, such as microcephaly, a “carp-shaped” mouth, deeply set eyes, prominent ears, and/or midfacial hypoplasia. Some affected individuals may also have visual abnormalities, hearing impairment, genital malformations, structural heart defects, and/or other physical abnormalities	unkn	
**19p13.12 DS**	0.8 Mb chr19: 15,328,527–16,092,461	Mild HL	occ	Defects of the branchial arches (preauricular tags, ear canal stenosis), mild ID	unkn	[9]
**Chr 22q11.2 DS, Distal (OMIM #611867) DiGeorge and velocardiofacial Distal S (DGS/VCFS Distal S)**	less than 3 Mb on distal Chr 22q11.2, between LCR22-4 and LCR22-6	HL, sensorineural and/or conductive, unilateral	**com**	History of prematurity, pre- and postnatal growth retardation, DD; slight dysmorphic features: arched eyebrows, deep-set eyes, broad nose, hypoplastic alae nasi, smooth philtrum, down-turned mouth, micrognathia, pointed chin	unkn	
**Chr 22q13.33 DS (OMIM #606232)** **Phelan-McDermid S**	160 kb to 9 Mb (*SHANK3*)	HL	less than 20% of patients, important in those with ring Chr 22, who are at risk for NF2	Neonatal hypotonia, absent to severely delayed speech, DD, minor dysmorphic facial features. Most affected individuals have moderate-to-profound ID. Other features include large fleshy hands, dysplastic toenails, and decreased perspiration that results in a tendency to overheat. Normal stature and normal head size distinguish Phelan-McDermid syndrome from other autosomal Chr disorders. Behavior characteristics include mouthing or chewing non-food items, decreased perception of pain, and autism spectrum disorder or autistic-like affect and behavior	unkn	
**Chr Xp11.3 microdeletion (OMIM #300578)**	ChrX:42,500,001–47,600,000	HL	occ	Bilateral, closed funnel retinal detachments consistent with a clinical diagnosis of Norrie disease, CHD, dysmorphic facies	*NDP* and *KDM6A*	[33]
**Chr Xq21 DS (OMIM # 303100I Choroideremia, deafness, and mental retardation**	Dels including at least *CHM* and *POU3F4*	SNHL, conductive, mixed, HL. Progressive, profound HL. Inner ear malformation (IP3)	**very com**	In males, choroideremia (progressive nyctalopia and eventual central blindness; onset in second to third decade), deafness (sensorineural and conductive) with inner ear malformation and DD/ID. Female carriers are generally asymptomatic, but they may show mild signs of choroideremia and rarely mild HL and inner ear malformation.	*POU3F4* °	
**Chr Xq22.1 DS** **(OMIM #304700) Mohr-Tranebjaerg S (MTS)**	dels in Xq22 (*DDP* and *BTK*)	SNHL, prelingual or postlingual, progressive	**com**	Deafness-Dystonia-Optic Atrophy/Neuronopathy Syndrome (DDP): hearing impairment in early childhood, slow progressive dystonia or ataxia in teens, slow progressive decreased visual acuity from optic atrophy beginning at approximately 20 years old, dementia beginning at approximately age 40 years. Psychiatric symptoms, such as personality changes and paranoia appear in childhood and progress. Females may have mild hearing impairment and focal dystonia. X linked agammaglobulinemia (XLA) if *BTK* is included in the Del	*TIMM8A*	[34,35,36]
**Chr Xq22.3 centromeric DS (OMIM #308940)** **Alport S and diffuse leiomyomatosis (ATS-DL)**	Dels involve *COL4A5* and extend centromerically to the gene ‘X-linked semi-dominant’ inheritance	bilateral high-tone SNHL; moderate; SNHL	**com**	Diffuse leiomyomatosis with Alport syndrome	*COL4A5* °	
**Chr Xq22.3 telomeric DS (OMIM #300194)** **Alport S, with intellectual disability, ATS-ID, AMME complex**	Dels involve *COL4A5* and extend telomerically to the gene. The CR for ID is of 380 kb~, containing *FACL4* (*ACSL4*), *NXT2*, *KCNE5* (*KCNE1L*), *GUCY2F*	SNHL	**com**	Hematuria, renal failure, HL, NDD, midface retrusion, elliptocytosis	*COL4A5* °	
** Chr Xq28 DS ** **(OMIM #300475)** **Deafness, dystonia, and cerebral hypomyelination (DDCH) Contiguous *ABCD1*/*DXS1375E* Deletion Syndrome, Included (CADDS)**	*BCAP31* is flanked by *SLC6A8* and *ABCD1*. Patients with dels including *BCAP31* and *SLC6A8* have the same phenotype as *BCAP31* patients. Patients with dels of *BCAP31* and *ABCD1* have contiguous *ABCD1* and *DXS1375E*/*BCAP31* deletion syndrome (CADDS), and demonstrate a more severe neurological phenotype with cholestatic liver disease and early death	SNHL	**com**	Severe ID, dystonia, cerebral hypomyelination. Female carriers are mostly asymptomatic but may present with deafness and/or LD, ID	*BCAP31*	[37]

Del: deletion; Dup: duplication; S: syndrome; DS: deletion syndrome; Chr: chromosome; bkp: breakpoint; MCR: minimal chromosome region; SRO: small region of overlap; HL: hearing loss; SNHL: sensorineural hearing loss; HA: hyperacusis; auditory neuropathy: AN; occ: occasional; com: common; ID: intellectual disability; DD: developmental delay; LD: language delay; ASD: autism spectrum disorder; ADHD: attention deficit hyperactivity disorder; CHD: congenital heart defect; IUGR: intrauterine growth restriction; unkn: unknown; hyp: hypothesized; Ref: references; *: other than those in OMIM; °: included in hereditaryhearingloss.org (accessed on 29 April 2024) [38].

**Table 2 genes-15-00677-t002:** Microduplication syndromes with HL.

Duplication Syndrome (DupS)	Deletion Size, Genomic Coordinates (GRCh38)	HL Classification	HL Frequency	Phenotype	Gene/s Associated with HL °
**Chr 5p13 DupS** **(OMIM #613174)**	Dup of 0.25 to 1.08 Mb encompassing *NIPBL* chr5:28,900,001–42,500,000	H Disorder	occ	DD and LD, behavioral problems and peculiar facial dysmorphisms	unkn
** 8q12.2 ** **DupS**	Dup MCR 1.2–1.6 Mb (*CA8*, *RAB2*, *RLBP1L1*, *CHD7*) Entire *CHD7* del does not lead to CHARGE	HL, Mondini malformation, malformation of the ear canal	**com**	Hypotonia, failure to thrive, ID, Duane anomaly, ASD, facial features, with or without heart defects	*CHD7* °
**Chr 9q21.11 DupS_(OMIM #613558) Deafness autosomal dominant 52; DFNA51**	Dup of 269 kb (hg19) chr9:42,840,853–69,663,410	HL, age-related	com (a single family reported)	Adult-onset, progressive nonsyndromic HL with onset in the fourth decade, first affecting high frequencies and later becoming severe to profound at all frequencies. No evidence of vertigo, dizziness, disequilibrium, or imbalance.	*TJP2/ZO-2* ° hyp
**Chr 10q24 DS (OMIM #246560) Split-hand/foot malformation 3**	Dup of at least 325 kb MCR dup including only *BTRC* and *POLL* chr10:95,300,001–104,000,000	HL, conductive or mixed from chronic serous otitis media	**com**	Split-hand/split-foot malformation (SHFM3). Some patients exhibit ID, ectodermal and craniofacial findings, orofacial clefting	unkn

Del: deletion; Dup: duplication; S: syndrome; DupS: duplication syndrome; Chr: chromosome; HL: hearing loss; SNHL: sensorineural hearing loss; occ: occasional; com: common; ID: intellectual disability; DD: developmental delay; LD: language delay; ASD: autism spectrum disorder; ADHD: attention deficit hyperactivity disorder; CHD: congenital heart defect; IUGR: intrauterine growth restriction; unkn: unknown; hyp: hypothesized; °: included in hereditaryhearingloss.org (accessed on 29 April 2024) [38].

**Table 3 genes-15-00677-t003:** Reciprocal microdeletion and microduplication syndromes with HL.

Reciprocal Deletion/Duplication Syndrome (DELS/DUPS)	Del/Dup Size, Genomic Coordinates (GRCh38)	HL Classification	HL Frequency	Phenotype	Gene/s Associated with HL °	Ref
**Chr 7q11.23 DELS, distal, 1.2Mb (OMIM #613729)**	Del of 1.5 to 1.8 Mb Chr7:72,700,001–77,900,000	SNHL, mild to moderate, high-frequency, progressive; Chronic otitis media/possible mixed HL; specific phobias for certain sounds; Hyperacusis associated with absence of contralateral acoustic reflexes	>60% >90% (adults)	DD, ID (usually mild), a specific cognitive profile, unique personality characteristics, cardiovascular disease (supravalvar aortic stenosis, peripheral pulmonary stenosis, hypertension), connective tissue abnormalities, growth deficiency, endocrine abnormalities (early puberty, hypercalcemia, hypercalciuria, hypothyroidism), and distinctive facies. Hypotonia and hyperextensible joints can result in delayed attainment of motor milestones. Feeding difficulties often lead to poor weight gain in infancy	*ELN* hyp	[39]
**Chr 7q11.23 DUPS (OMIM #609757)** **Williams-Beuren Region DUPS**	Dup of 1.5 to 1.8 Mb Chr7:72,700,001–77,900,000	HL	~5%	Severe impairment in expressive language, including a phonologic disorder, delayed motor and social skills, neurologic abnormalities, behavior issues including anxiety disorders (especially social anxiety disorder, social phobia), selective mutism, ADHD, oppositional disorders, physical aggression, autism spectrum disorder. ID in some individuals. Distinctive facial features are common. Cardiovascular disease includes dilatation of the ascending aorta	*ELN* hyp	
**Chr 7q21.3 DELS/DUPS (OMIM #183600) Split-Hand/Foot Malformation with or without deafness/Split-Hand/Foot Deformity 1**	Del, dup, or rearrangement involving DSS1, DLX5, DLX6, and possible regulatory elements in the region	SNHL. Mixed HL	occ	Limb malformation presenting with syndactyly, median clefts of the hands and feet, and aplasia and/or hypoplasia of the phalanges, metacarpals, and metatarsals. Some patients have also mental retardation, ectodermal and craniofacial findings, orofacial clefting, and SNHL	unkn	[40,41]
**Chr 10p14** **DELS/DUPS (OMIM #146255)** **HDRS/Barakat S**	Del, dup involving *GATA3*	Early onset, moderate-to-severe SNHL, typically bilateral	very com	Hypoparathyroidism (H), nerve deafness (D) and/or renal disease (R). Variable clinical features include hypogonadotrophic hypogonadism, polycystic ovaries, CHD, RP, ID	*GATA3* °	[42]
** 11p15.5 ** **Beckwith-Wiedemann S (BWS, OMIM #130650)/Russell-Silver S (RSS, OMIM #180860)**	Del/dup of imprinted region of 11p15.5	Conductive HL due to fixation of the stapes (BWS) SNHL, cochlear malformation (RSS)	rare	BWS is a growth disorder variably characterized by macroglossia, hemihyperplasia, omphalocele, neonatal hypoglycemia, macrosomia, embryonal tumors, visceromegaly, adrenocortical cytomegaly, kidney abnormalities, and ear creases/posterior helical ear pits. Adult heights are typically within the normal range. RSS is typically characterized by asymmetric gestational growth restriction resulting in SGA, with relative macrocephaly at birth, prominent forehead with frontal bossing, and frequently body asymmetry. Postnatal growth failure, and in some cases progressive limb length discrepancy and feeding difficulties. Additional clinical features include triangular facies, fifth-finger clinodactyly, and micrognathia with narrow chin. Adult height in untreated individuals is ~3.1 ± 1.4 SD below the mean	unkn	[43,44]
**Chr 16p12.2-p11.2 DELS (OMIM #613604)**	del of 7.1 to 8.7 Mb SRO of 7.1 Mb chr16:21,200,001–35,300,000	HL uncommon. Frequent ear infections with potential conductive HL. Mixed HL	rare	Facial anomalies (flat faces, downslanting palpebral fissures, low-set and malformed ears, and eye anomalies), feeding difficulties, DD, significant delay in speech development, ID, and recurrent ear infections	*OTOA* ° hyp	[45]
**Chr 16p12.2-p11.2 DUPS**	dup of 7.1 to 8.7 Mb	HA	rare (HA)	ASD	*OTOA* ° hyp	[46]
**Chr 17p11.2 DELS (OMIM #182290)** **Smith-Magenis S**	3.7 Mb interstitial del	HL, usually mild and related to chronic and recurrent otitis media, Eustachian tube dysfunction and craniofacial anomalies	com (more com in dels than in SNV in *RAI1*)	Feeding difficulties, failure to thrive, hypotonia, hyporeflexia, lethargy, sleep disturbance, distinctive physical features (coarse facial features progressing with age), DD, ID, specific behavioral abnormalities/maladaptive behaviors, childhood-onset abdominal obesity	Craniofacial anomalies and infections *MYO15A* °	[47,48]
**Chr 17p11.2 DUPS (OMIM #610883)** **Potocki-Lupski S**	3.7 Mb interstitial dup	Mild high-frequency SNHL; Mild sensitivity loss at 4000 Hz.	rare	DD, ID, behavioral abnormalities, ASD, hypotonia, oropharyngeal dysphagia leading to failure to thrive, CHD, hypoglycemia, growth hormone deficiency, and mildly dysmorphic facial features	unkn	[49]
**Chr 22q11.2 DEL (OMIM #188400)** **DiGeorge/velocardiofacial S (DGS/VCFS)**	1.5 to 3.0 Mb del Frequent 2.54-Mb del (>40 genes); atypical, “nested” del; 1.5-Mb del LCRs A–B; del LCRs A–C; del LCRs B–D or C–D.	SNHL and/or conductive HL inner ear anomalies (semicircular canal, ossicular, vestibular aqueduct, and vestibular anomalies)	very com	CHD, conotruncal malformations, palatal abnormalities (velopharyngeal incompetence, submucosal cleft palate, bifid uvula, and cleft palate), hypocalcemia from parathyroid/thymic hypoplasia, immune deficiency, characteristic facial features, micrognathia, ear abnormalities, DD and learning difficulties, short stature, laryngotracheoesophageal, gastrointestinal, ophthalmologic, central nervous system, skeletal, and genitourinary anomalies, psychiatric illness, and autoimmune disorders. Highly variable phenotype	*TBX1* cranio-facial anomalies and immunologic factors, inner ear malformation	[50]
**Chr 22q11.2 DUP (OMIM # 608363)**	1.5 to 3.0 Mb proximal tandem dup	HL, mostly conductive, secondary to recurrent otitis media; mixed HL	com	Learning disability, ID, DD, growth retardation, muscular hypotonia, dysmorphic features, CHD, visual and hearing impairment, seizures, microcephaly, ptosis, and urogenital abnormalities. Highly variable phenotype, ranging from asymptomatic to severe	unkn	

Del: deletion; Dup: duplication; S: syndrome; Chr: chromosome; HL: hearing loss; SNHL: sensorineural hearing loss; occ: occasional; com: common; ID: intellectual disability; DD: developmental delay; LD: language delay; ASD: autism spectrum disorder; ADHD: attention deficit hyperactivity disorder; CHD: congenital heart defect; IUGR: intrauterine growth restriction; RP: retinitis pigmentosa; unkn: unknown; hyp: hypothesized; °: included in hereditaryhearingloss.org (accessed on 29 April 2024) [38].

## Data Availability

Data are contained within the article or Appendix A.

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
