# Peer review of "Contiguous Gene Syndromes and Hearing Loss: A Clinical Report of Xq21 Deletion and Comprehensive Literature Review"

_genes, 2024, doi:10.3390/genes15060677_

Round 1

Reviewer 1 Report

Comments and Suggestions for Authors

Summary: The authors describe a case report of an individual with hearing loss with a contiguous gene deletion on chromosome X. They then review the literature of all individuals with hearing loss who have contiguous gene deletions and duplications.

Major points:

-I would argue that what the authors are calling HL contiguous deletion syndromes are often referred to by the other features of the condition. Even in the case report they are detailing from their clinic, I would argue that this is an individual with neurodevelopmental disorder who also happens to have hearing loss. It is common that hearing loss can be a feature of more severe syndromic conditions.

-Isolating the HL from the other features in the contiguous gene del and dup table is slightly incorrect. For example, in the Chr Xq28 DS where BCAP31 is listed as the causative gene for the HL. If you go to OMIM or GenCC, this gene is listed as definitively associated with Deafness, dystonia, and cerebral hypomyelination. I think this type of review is missing the point for both clinicians and laboratorians. Yes, they should be aware that these dels or dups can also cause HL along with a range of other severe features.

Minor points:

-The dashes in lines 30-33 on page one are distracting. I would suggest removing them.

-Line 50, page 2 I would change variant to variant(s) as a majority of single gene HL disorders are autosomal recessive and your statement makes it sound like only a single variant is required.

-Line 64, page 2, I would change to young boy (if he came to you at 3yrs of age), newborn boy if he came to you at birth

-The authors need to add DS to the abbreviation footnotes in table 1

Comments on the Quality of English Language

-Some of the syntax reads strangely. Please have a native English speaker read through the manuscript and suggest some edits. For example, taking this sentence from the abstract “Giving the crucial role of personalized management and treatment of hearing loss (HL), etiological investigations are early performed, and genetic analysis significantly contribute to the determination of most syndromic and nonsyndromic HL.” would sound better as  “Given the crucial role of personalized management and treatment of haring loss, etiological investigations are performed early, and genetic analysis significantly contributes to the determination of most syndromic and nonsyndromic HL.”

Author Response

Regarding reviewer’s major point 1, “I would argue that what the authors are calling HL contiguous deletion syndromes are often referred to by the other features of the condition. Even in the case report they are detailing from their clinic, I would argue that this is an individual with neurodevelopmental disorder who also happens to have hearing loss. It is common that hearing loss can be a feature of more severe syndromic conditions.”

We agree that we focus on syndromic conditions in which HL is generally one of the clinical features. In the introduction, we replaced ‘HL contiguous gene syndromes’, that is incorrect/misleading, with ‘Contiguous gene syndromes exhibiting HL in association with congenital anomalies, neurodevelopmental disorders, and various syndromic conditions’.

We then emphasized the importance of HL as the possible earlier sign of a CGS, as it occurred to the patient reported by us. Indeed, thanks to the newborn hearing screening testing, HL is identified at birth, and treated, according to JCIH recommendations within the first 3-6 months of life.

Regarding reviewer’s major point 2, “Isolating the HL from the other features in the contiguous gene del and dup table is slightly incorrect. For example, in the Chr Xq28 DS where BCAP31 is listed as the causative gene for the HL. If you go to OMIM or GenCC, this gene is listed as definitively associated with Deafness, dystonia, and cerebral hypomyelination. I think this type of review is missing the point for both clinicians and laboratorians. Yes, they should be aware that these dels or dups can also cause HL along with a range of other severe features.”

One of the major aims of our review was to investigate, estimate, and characterize the presence of HL within described CGS. Tables 1a-c list CGS in which HL is or may be an important clinical feature, briefly detailing characteristics of HL. Genes responsible or possibly candidate for HL are reported according to updated literature. In addition, we indicate genes in HL-database (hereditaryhearingloss.org).  Some of these genes, such as BCAP31, have a pleiotropic effect.

Considering major points, in the light of better focusing on the messages, we removed the repetitions and the unnecessarily redundant sentences. Occasionally we have rearranged the order of the sentences to make the to make the messages more accessible to the reader.

We accepted also all minor revisions suggested by both reviewers.

As for the Quality of English Language, the text has been read and corrected by a native English speaker, as suggested.

Reviewer 2 Report

Comments and Suggestions for Authors

This is a interesting article reporting a new clinical report of a young boy with Xq21 deletion syndrome, as well as a comprehensive litterature review about hearing loss associated with contiguous gene microdeletion / microduplication syndromes. This is a very interesting and instructive article.

I have only minor comments.

- Abstract: I suppose that “familiars” should be most likely replaced by “families”.

- Line 197: DFN3 refers to the former nomenclature. DFNX2 is now preferred.

- Overall, the number of abbreviations should be kept minimal. However, where necessary, please define the abbreviation on first use, and then use it throughout the manuscript. For example, incomplete partition type III is abbreviated to “IP3” line 113, and again line 328.

- The manuscript should be checked to correct some minor grammatical and typo errors. For example line 382 “POU3F4-realted” should be replaced by “POU3F4-related”.

Comments on the Quality of English Language

Overall, the quality of English language is good. Some minor editing is needed.

Author Response

We accepted also all minor revisions suggested by both reviewers.

As for the Quality of English Language, the text has been read and corrected by a native English speaker, as suggested.

Round 2

Reviewer 1 Report

Comments and Suggestions for Authors

I think that the authors have better framed this article by mentioning that HL is often part of a constellation of other features.